# Fault Segment Location for MV Distribution System Based on the Characteristic Voltage of LV Side

**Dehai Zhang, Wenhai Zhang \*, Changzhi Wang and Xianyong Xiao**

College of Electrical Engineering, Sichuan University, Chengdu 610052, China
\* Correspondence: zhangwh2079@scu.edu.cn

**Abstract:** The voltage and current measurement of the medium-voltage (MV) side are used for the traditional fault location method, which leads to a high installation cost of the measurement and complicated post-operational and maintenance work. A fault location method is proposed based on the distributed measurement at the low-voltage (LV) side. On the analysis of voltage distribution rules and the influence of the distribution transformer on voltage transmission, obvious differences are found in the variation of voltage (phase voltage and sequence voltage) at the LV side for different faults—specifically, the detection sensitivity of the voltage to different faults varies. Therefore, a fault location method for the MV distribution network based on characteristic voltage at the LV side is proposed. Firstly, the characteristic voltage is selected adaptively according to the fault type. The suspected fault path is then determined by the characteristic voltage amplitude of measuring points. Finally, the fault segment is located via the characteristic current of each segment. This method can be applied in combination with the widely used LV measurement equipment such as the power consumption information acquisition system and the intelligent fusion terminal to acquire LV side voltage without adding new measurement devices. The distribution difference of the characteristic voltage at the LV side is applied for accurate fault finding, which is an economical and practical new idea for the fault location of the distribution network. The correctness and effectiveness of the method were verified by the simulation of the IEEE 34 system built in PSCAD/EMTDC and a real distribution network.

**Keywords:** characteristic voltage; distributed low voltage side voltage; fault location; medium voltage distribution network; segment location





## 1. Introduction

As an important part of linking the power system and customers, the Medium-Voltage (MV) distribution network directly supplies power to customers. Its operational reliability directly affects the continuity of the power supply. Faults are difficult to avoid due to the complex environment and wide distribution range of the MV distribution network. Statistically, about 90% of power outages are caused by faults in the distribution network. The fast and accurate location of the fault point is critical for reducing the outage time and loss as well as improving the power supply reliability of the grid.

Numerous studies have been conducted for fault location in MV distribution networks. According to the difference of location principle and measuring points, the location methods can be classified into the following three types: (1) the impedance method, which is based on the station-side voltage or current measurement; (2) the traveling wave method, which is based on the station-side or multi-point measurement; (3) the wide-area communication method. Among them, the impedance [1–4] and traveling wave methods [5–8] are traditional location methods and have been widely applied in fault location in the transmission network. However, many difficulties arise when these methods are applied to the distribution network. The impedance method fails to completely solve the problems caused by various line types and branches. The traveling wave method faces the challenge

of wavefront identification due to short lines and the complex structure of distribution network. Moreover, there are practical application challenges, such as a high sampling frequency and significant investments in equipment.

Increasing information has recently been provided for the fault location of the distribution network with the diversification and generalization of the measurement device. To overcome the difficulties caused by the complex structure of the distribution network for fault location, the method based on wide area communication has become a popular research topic [9–14]. The measurement objects typically include feeder currents and node voltages. Meanwhile, a synchronous phasor measurement has also been applied in the distribution network's fault location [15–19]. The method of the feeder current measurement is based on the traditional distribution network automation and fault indicator, for which the location accuracy is proportional to the number of monitoring terminals. This type of method requires the installation of current measurement devices on feeders, which requires a significant amount of installation, operation, and maintenance. Node voltage measurement-based methods usually perform voltage measurements at switching stations, ring network cabinets, and distribution transformers to realize fault location based on a sparse voltage measurement on the MV side. Currently, for a multi-point quantitative location, it is more common to introduce a current injection source at the fault point [20–23]. The voltage of the sparse measurement and node impedance matrix are used to calculate the node injection current, which reflects the fault segment. In [24], an impedance-based method is used to derive multiple fault points. The sparse measurement data are employed to establish LV zones to exclude the pseudo-fault point. In [25], the relevant parameters are improved on this basis to make the location results more accurate. In addition, the direct voltage matching method has also been used to perform multi-point measurements [26,27]. However, fault cases must be simulated for all nodes, which requires high accuracy for grid modeling and a complicated process. The fault resistance must be estimated by an iterative method, for which the accuracy is difficult to guarantee. In [28], the property of zero reactive power consumed by the resistive fault is used to establish the relevant equations to determine the fault distance. However, it may generate pseudo-fault points, which need to be further excluded. In [29], the sparse voltage measurements are used to calculate the fault current when all nodes subsequently fail. Faults can be found in accordance with the principle of minimum current error, which could also generate pseudo-fault points.

The above sparse-measurement based location methods are mainly based on MV side voltage measurements for fault location, which share common limitations. Although MV side voltage information is a more intuitive and accurate reflection of fault information on MV lines, it requires the installation of additional voltage transformers, which increases costs and may also pose a risk of ferro-resonance to the system. In a complex and large distribution network, the MV side requires a large number of measurement points to achieve accurate positioning, which greatly increase the cost of positioning and are difficult to apply in practical engineering. For this reason, a practical solution for the distribution network fault location based on the LV side voltage measurement is presented in this paper. In [30], the negative sequence voltage changes at each measuring point on the LV side before and after the fault is divided into different groups, and the group with the maximum mean value is used to determine the fault location. This method may yield incorrect results for network with long laterals, and it cannot be used for the location of the three-phase fault because negative sequence voltage is selected as the single analysis object for all fault types. Voltage similarity matching has been previously used for fault location based on LV side measurements [31]. Moreover, a single negative sequence component was used for the analysis, which cannot be used for locating three-phase ground faults. In [32], three-phase voltage sags caused by short-time connection of the auxiliary resistor are used for fault segment identification. However, the operation of the auxiliary resistor will increase the complexity of the overall location scheme and the increased fault current will pose a threat to line insulation. A location method has been proposed based on the ratio of positive and negative sequence voltages with a limited number of measuring points

on the LV side [33]. Most existing fault location methods based on the LV side measurement employ negative sequence components as their analysis object, which cannot be used for a three-phase fault. Some location schemes may receive multiple or incorrect results for a complex network with numerous and long laterals. In addition, the same characteristic quantity has a different sensitivity to different faults. Single negative sequence voltage does not give the best location results in some cases.

Compared to the MV side, it is relatively convenient to obtain electrical quantities on the LV side. LV side measurements can be combined with terminals such as intelligent fusion terminals, electricity consumption information acquisition systems, and smart meters to collect relevant data. The operation, maintenance costs, and hardware investments of LV side measurement devices are relatively low. In view of the problems of the above MV side location methods, fault segment location based on LV side characteristic quantities is considered. Based on analysis of the distribution characteristics of each voltage on MV side and the voltage transmission characteristics of distribution transformers, significant differences were found in the influence of the various faults on the voltage characteristics of the LV side. In a single-phase fault, the phase voltage and negative sequence voltage changes on the LV side are small, but the negative sequence voltage maximizes at the fault point. For a phase-to-phase fault, the LV side phase voltage changes significantly corresponding to the two faulty phases. Therefore, the characteristic voltages of different faults were determined based on the fault distribution characteristics of each voltage quantity on the LV side. The characteristic quantity was selected adaptively according to the fault type, which is conducive to the most sensitive reflection of the fault location. The characteristic voltage is used to determine the faulty path and the fault segment search algorithm to avoid the misjudgment. These two principles were combined to achieve the fault segment location of the MV distribution network based on the characteristic voltage measurement on the LV side. Compared with the previous method based on the MV measurement, this method does not need new measuring equipment but can use the existing low-voltage measuring equipment in the distribution network, such as the electricity information acquisition system and the intelligent fusion terminal, to collect the low-voltage side voltage data, which greatly reduces the positioning cost. Compared with the previous method based on the LV side, the proposed method in this study can realize the fault path and fault segment identification only by using the voltage value of the low voltage side measuring point without particularly complicated mathematical calculation and procedures, and the implementation is simple. According to the fault type, the characteristic voltage for the segment location can be determined adaptively so that all short-circuit faults can be located. The section search algorithm is used to avoid the generation of pseudo fault points. The contribution of this work can be summarized as below:

1.  The fault distribution characteristics of the LV side are analyzed and used for fault location.
2.  The transmission characteristics of the distribution transformer to the voltage on both sides of the MV and LV are analyzed, which is the basis of reflecting the fault location of the medium voltage line from the characteristic quantity on the low voltage side.
3.  The sensitivity of different voltages to different faults is analyzed and the characteristic voltages of each fault type are determined accordingly.
4.  A method is proposed to determine the fault path and fault section by using the characteristic voltage distribution characteristics of the LV side, which does not require additional measurement equipment, greatly reduces the positioning cost, and is simple to calculate.
5.  This method can be combined with existing MV side-based location methods to achieve more economical and accurate fault location.

The remainder of this paper is organized as follows. In Section 2, the distribution characteristics of the voltage on the MV and LV sides is analyzed. In Section 3, the principle and steps of the proposed fault location method are described in detail, including the configuration of measuring points. In Section 4, the performance of the fault location

method is evaluated via the two distribution network models. Conclusions are provided in the last part.

## 2. Fault Voltage Distribution Characteristics Analysis

When a fault occurs in the MV distribution network, the voltage at the fault point is significantly reduced and the voltage amplitude of the entire network changes to a certain extent. Hence, the distribution characteristics of the system voltage can be used to locate the fault in the network. The LV side measurement information can indirectly reflect the MV side voltage information owing to the characteristics of transformer transmission. However, the fault information of the MV side cannot be fully transmitted to the LV side, owing to the limitation of the connection mode of the distribution transformer windings. In this section, the distribution law of the LV side voltage under different faults will be discussed in accordance with the transformer transmission characteristics based on the distribution law of the MV side voltage in the entire network.

### 2.1. Study of Fault Voltage Distribution Law on MV Side

Figure 1 shows the fault sequence network diagram when a single-phase fault occurs in the effective grounding system, where $Z_{s1}$, $Z_{s2}$, and $Z_{s0}$ denote the positive, negative, and zero sequence impedance of the system; $V_{1f}$, $V_{2f}$, and $V_{0f}$ denote the positive, negative, and zero sequence voltages at the fault point; and F indicates the fault point. The shade of the red line in the figure indicates the magnitude of the sequence voltage, which can be used to better understand the distribution law of positive, negative, and zero sequence voltages. The amplitude of the positive sequence voltage gradually decreases on the fault path due to the large fault current flowing from the source side of the system to the fault point. The voltage variation of branches is small because only the load current flows through it. The amplitude of the positive sequence voltage downstream of the fault point is close to the fault point because the load current is small. The distribution law of negative and zero sequence voltages on the faulty path is opposite to that of the positive sequence voltage. Its magnitude gradually increases from the source end to the fault point with a maximum value at the fault point, and the voltage downstream is close to the fault point with a larger magnitude. Notably, the difference in the distribution of the voltage in the network is essentially the voltage drop generated by the current on the line. The overall distribution law of the sequence voltage under a different fault is consistent, although the level sequence current significantly varies under the same law of the sequence voltage distribution. Hence, the difference in the distribution of each voltage quantity in the network under different faults is also different.

Unlike the positive, negative, and zero sequence network, which can be completely decoupled, the distribution law of the phase voltage and line voltage is more complex. For simplification, only the fault phase voltage and the line voltage associated with the fault phase are analyzed in this study. The distribution law of the fault phase voltage is similar to that of positive sequence voltage. Its amplitude decreases gradually from the source side of the system to the end of the fault downstream, and the voltage variation per unit line length downstream of the fault is significantly small. It should be noted that the variation speed of the voltage on the line is positively related to the current amplitude. The distribution law of line voltage is directly related to the system neutral grounding mode and fault type. For a single-phase fault in the small current grounding system, the LV side line voltage is almost unaffected. For the same fault in the small resistance grounding system, the line voltage reduces gradually from the source end to the fault point; for phase-to-phase fault, the line voltage reduces gradually to zero from the source end to the fault point, and the line voltage on the downstream side of the fault is close to the fault point.

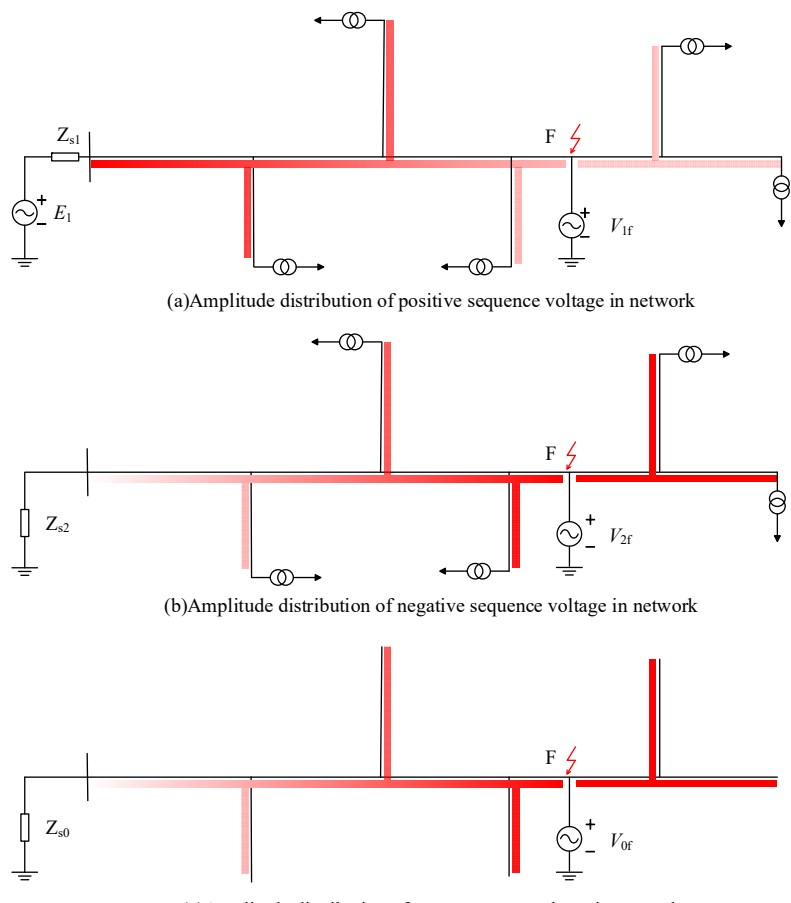

(a)Amplitude distribution of positive sequence voltage in network

(b)Amplitude distribution of negative sequence voltage in network

(c)Amplitude distribution of zero sequence voltage in network

**Figure 1.** Sequence voltage distribution law of single-phase ground fault.

Theoretical calculations and simulation verification were conducted to further compare the variation characteristics and distribution law of voltages under different fault types and neutral grounding modes. The possible variations of each voltage at the fault point before and after the fault are presented in Table 1, where $V_1$, $V_2$, and $V_0$ denote positive, negative, and zero sequence voltages, and $V_P$ and $V_L$ denote phase and line voltages, respectively. Table 1 demonstrates that the variation characteristics of each voltage under different fault types vary greatly. When selecting the voltage distribution for the fault location, it is necessary to combine the neutral grounding mode of the system and the fault type to adaptively select the characteristics voltage with the largest change in magnitude and the largest difference in distribution for location. The neutral point ungrounded system and the neutral point directly grounded system are two extreme cases, where the voltage variations can be directly calculated. The results are shown in Table 1, where LG, LL, LLG, and 3L represent, respectively, the single-phase, phase to phase, two-phase to ground, and three-phase fault.

**Table 1.** Voltage variation of MV side at fault point under different faults.

| Fault Type | Voltage | Voltage Variation (p.u.) | | |
|---|---|---|---|---|
| | | Directly Grounded | Grounded by Low Resistance | Grounded by Arc Supression Coil and Ungrounded |
| LG | $V_1$ | 0.2 | 0~0.2 | 0 |
| | $V_2$ | 0.2 | 0~0.2 | 0 |
| | $V_0$ | 0.6 | 0~0.6 | 1 |
| | $V_P$ | 1 | 1 | 1 |
| | $V_L$ | 0.28 | 0~0.28 | 0 |

**Table 1.** *Cont.*

| Fault Type | Voltage | Voltage Variation (p.u.) | | |
| --- | --- | --- | --- | --- |
| | | Directly Grounded | Grounded by Low Resistance | Grounded by Arc Supression Coil and Ungrounded |
| LL | $V_1$ | 0.5 | 0.5 | 0.5 |
| | $V_2$ | 0.5 | 0.5 | 0.5 |
| | $V_0$ | 0 | 0 | 0 |
| | $V_P$ | 0.5 | 0.5 | 0.5 |
| | $V_L$ | 1 | 1 | 1 |
| LLG | $V_1$ | 0.57 | 0.5~0.57 | 0.5 |
| | $V_2$ | 0.43 | 0.43~0.5 | 0.5 |
| | $V_0$ | 0.43 | 0.43~0.5 | 0.5 |
| | $V_P$ | 1 | 1 | 1 |
| | $V_L$ | 1 | 1 | 1 |
| 3L | $V_1$ | 1 | 1 | 1 |
| | $V_2$ | 0 | 0 | 0 |
| | $V_0$ | 0 | 0 | 0 |
| | $V_P$ | 1 | 1 | 1 |
| | $V_L$ | 1 | 1 | 1 |

Due to the influence of system parameters and neutral grounding impedance (arc suppression coil, grounding resistance), it is difficult to accurately calculate the voltage variation in small resistance grounding and resonate grounding systems. Assuming that the arc suppression coil in the resonant grounding system is fully compensated, its typical voltage variation is consistent with the ungrounded system. In the small resistance (10 Ω) grounding system, the voltage change is between the direct grounding and the ungrounded system due to the influence of the fault distance. It should be noted that the above analysis only considers the voltage variation law on the 10 kV side of the distribution network and the MV side sequence voltage information cannot be fully transmitted to the LV side due to the influence of the distribution transformer. The voltage variation on the LV side needs to be analyzed in combination with the distribution transformer.

*2.2. Effect of the Distribution Transformer on Voltage LV Side Characteristics*

To further obtain the changing characteristics and distribution laws of the LV side voltage under different fault conditions, the voltage characteristics of the LV side are analyzed with the operational characteristics of the distribution transformers in this section. The 10/0.4 kV distribution transformer in China primarily adopts the Dyn11 and Yyn0 coupling methods, and the typical coupling method Dyn11 is used as an example to reveal the transmission law of each voltage quantity. Based on the transfer characteristics of the transformer, the phase voltage on the LV side can be expressed as

$$\begin{bmatrix} U'_a \\ U'_b \\ U'_c \end{bmatrix} = \frac{1}{\sqrt{3}K} \begin{bmatrix} 1 & -1 & 0 \\ 0 & 1 & -1 \\ -1 & 0 & 1 \end{bmatrix} \begin{bmatrix} U_a \\ U_b \\ U_c \end{bmatrix} \tag{1}$$

where $U_a$, $U_b$ and $U_c$ denote the phase voltage at primary side (MV side); $U'_a$, $U'_b$ and $U'_c$ denote the phase voltage at secondary side (LV side); and $K$ is the transformer ratio. Based on the relationship formula, the LV side phase voltage corresponds to the MV side line voltage. Further, the LV side line voltage can be expressed as follows:

$$\begin{bmatrix} U'_{ab} \\ U'_{bc} \\ U'_{ca} \end{bmatrix} = \frac{1}{\sqrt{3}K} \begin{bmatrix} 1 & -2 & 1 \\ 1 & 1 & -2 \\ -2 & 1 & 1 \end{bmatrix} \begin{bmatrix} U_a \\ U_b \\ U_c \end{bmatrix} \tag{2}$$

where $U'_{ab}$, $U'_{bc}$ and $U'_{ca}$ indicate the line voltage of the secondary side. Compared to (1), the line voltage of the LV side is still a linear transformation of the MV side phase voltage, but the relationship is more complex.

To further analyze the variation law of the sequence voltage, the MV side positive, negative, and zero sequence are indicated by $U_1$, $U_2$, and $U_0$, and the corresponding sequence components of the LV side are indicated by $U'_1$, $U'_2$, and $U'_3$. The corresponding relationship between these sequence components of the medium and LV side is as follows:

$$\begin{bmatrix} U'_1 \\ U'_2 \\ U'_0 \end{bmatrix} = \frac{1}{K} \begin{bmatrix} e^{30°j} & 0 & 0 \\ 0 & e^{-30°j} & 0 \\ 0 & 0 & 0 \end{bmatrix} \begin{bmatrix} U_1 \\ U_2 \\ U_0 \end{bmatrix} \tag{3}$$

As shown in (3), the zero-sequence voltage cannot be transferred to the LV side. Both positive and negative sequence voltages are reduced in magnitude by the ratio when they are transferred, and their phase will be simultaneously shifted.

According to the transfer matrix of each voltage, the line voltage on the MV side corresponds to the phase voltage on the LV side, and, thus, the two are synchronized in terms of variation. For the sequence component, zero sequence cannot pass through the transformer, and positive and negative sequence components pass through the transformer and have corresponding changes in magnitude and phase, but the magnitude on both sides also stays synchronized in variation. The voltage changes on the LV side are shown in Table 2, where the meaning of each letter is the same as Table 1.

**Table 2.** Voltage variation of LV side at fault point under different faults.

| Fault Type | Voltage | Voltage Variation (p.u.) | | |
|---|---|---|---|---|
| | | **Directly Grounded** | **Grounded by Low Resistance** | **Grounded by Arc Supression Coil and Ungrounded** |
| LG | $V_1$ | 0.2 | 0~0.2 | 0 |
| | $V_2$ | 0.2 | 0~0.2 | 0 |
| | $V_0$ | 0 | 0 | 0 |
| | $V_P$ | 0.28 | 0~0.28 | 0 |
| | $V_L$ | 0.08 | 0~0.08 | 0 |
| LL | $V_1$ | 0.5 | 0.5 | 0.5 |
| | $V_2$ | 0.5 | 0.5 | 0.5 |
| | $V_0$ | 0 | 0 | 0 |
| | $V_P$ | 1 | 1 | 1 |
| | $V_L$ | 0.5 | 0.5 | 0.5 |
| LLG | $V_1$ | 0.57 | 0.5~0.57 | 0.5 |
| | $V_2$ | 0.43 | 0.43~0.5 | 0.5 |
| | $V_0$ | 0.43 | 0~0.43 | 0 |
| | $V_P$ | 1 | 1 | 1 |
| | $V_L$ | 0.57 | 0.5~0.57 | 0.5 |
| 3L | $V_1$ | 1 | 1 | 1 |
| | $V_2$ | 0 | 0 | 0 |
| | $V_0$ | 0 | 0 | 0 |
| | $V_P$ | 1 | 1 | 1 |
| | $V_L$ | 1 | 1 | 1 |

## 3. Fault Segment Location Method Based on LV Side Characteristic Voltage

### 3.1. Principle of Fault Segment Location

According to the analysis of the distribution characteristics of the fault voltage indicated in Section 2, the voltage variation range of the MV and LV sides varies for different faults. In order to realize the fault location based on the voltage distribution characteristics at the LV side, the voltage characteristic quantity with the largest distribution difference is selected as the characteristic voltage of this type of fault. In the case of a single-phase ground fault, the negative sequence component of the whole network reaches its maximum

value at the fault point. The fault segment can be determined by searching for the location of its maximum value. Similarly, the fault location can be determined by searching for the minimum value of phase voltage or the maximum value of negative sequence voltage for inter-phase faults. The characteristic voltage applicable to different faults will be provided based on data analysis in the following section. It should be noted that the measuring points often do not directly obtain the magnitude of the characteristic voltage at the fault point. Generally, the maximum value of the measured characteristic voltage is downstream of the fault. Namely, the fault is in the path between the measuring point corresponding to the maximum characteristic voltage and the substation. Once this path is determined, the fault segment can be identified by the difference between the upstream and downstream current of the fault.

### 3.2. Determination of the Characteristic Voltage

To locate the fault segment based on the distribution law of the characteristic voltage on the LV side, the characteristic voltage must satisfy the following two conditions. The characteristic voltage must have different magnitudes at the fault point compared to anywhere else in the network, such as the maximum or minimum value. The characteristic voltage differs significantly in the distribution law upstream and downstream of the fault and is easy to measure. Now, each voltage quantity on the LV side under different faults in Table 2 is analyzed according to the conditions. Only the negative sequence voltage in single-phase ground fault meets the condition at the same time. Both negative sequence voltage and phase voltage in the two-phase fault and the two-phase to the ground fault meet the conditions, but the phase voltage variation is larger. Both phase voltage and line voltage in the three-phase fault are available. It should be noted that the selection of the characteristic voltage will be affected by the connection mode of the distribution transformer. Since the distribution transformer primarily adopts the Dyn11 and Yyn0 connection modes, the selection of the characteristic voltage at the LV side is shown in Table 3.

**Table 3.** Determination of the characteristic voltage.

| Type of Distribution Transformer | LG | LL, LLG | 3L |
|---|---|---|---|
| Dyn11 | Negative voltage | Phase voltage | Phase voltage |
| Yyn0 | Negative voltage | Phase voltage | Phase voltage |

In an actual distribution network, to eliminate the negative sequence component introduced by the three-phase unbalance of the distribution network, the characteristic quantity of the negative sequence voltage for the single-phase fault location needs to be replaced by the change of the negative sequence voltage before and after the fault.

### 3.3. Suspicious Fault Path Determination and Segmentation

According to the characteristic voltage amplitude obtained from each measuring point, the measuring point is determined where the maximum or minimum characteristic voltage is located. The fault point can be initially determined to be located between the substation and the measuring point where the characteristic voltage optimum exists. This path is named the suspected fault path, where the fault may occur in any zone. As a simple distribution network topology shown in Figure 2, voltage measuring points are configured at the substation and the LV side at the end of each long branch (nodes 5, 7, 9, 11, and 14). When a single-phase ground fault occurs at F, as shown in Figure 2, the characteristic voltage magnitudes of the measuring points on the LV side of nodes 5, 7, 9, 11, and 14 are compared. The characteristic voltage (negative sequence voltage) magnitude on the LV side of node 11 is the largest, and the path between nodes 1 and 11 is identified as the suspicious fault path. Subsequently, the path is divided into several segments according to the branches with measuring points on the path. Nodes 2 and 3 have LV measuring points

at the end of the corresponding branches, so the suspicious fault path can be divided into three subsections and marked in order, as shown in the dashed box in Figure 2.

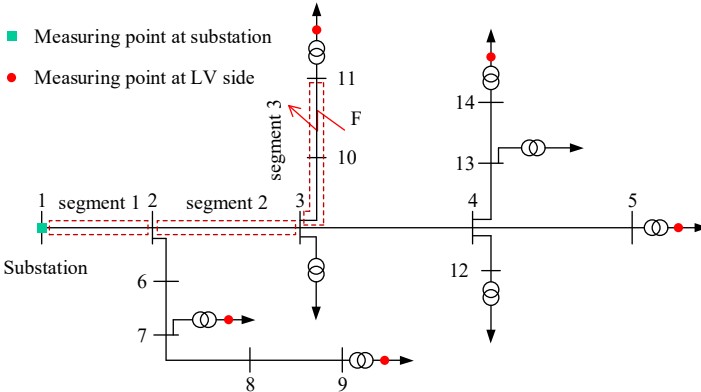

**Figure 2.** Schematic diagram of location method.

In practice, considering the measurement error of the equipment, there may be a situation in which the characteristic voltage of the fault point and its downstream branch measuring points are very close to each other. There will be more than one measuring point with the same maximum or minimum value of the characteristic voltage amplitude. In this scenario, the path between the first common node of these measuring points to the substation should be determined as the suspected fault path.

### 3.4. Fault Segment Search Algorithm

The difference in the distribution of the characteristic voltage in the grid is essentially the voltage drop on the line caused by the corresponding characteristic current and the larger characteristic current facilitating the fault location. Since the characteristic current above and below the fault tend to be considerably different, the characteristic current on the line can be calculated from the characteristic voltages for determining the fault section. A self-synchronization method of the fault data proposed in a previous study was adopted herein [34], which does not require strict synchronization requirements for the measurement device. The simplified schematic diagram of Figure 2 is shown in Figure 3. According to the assumptions in Section 3.3, the suspected fault path (node 1 to node 11) and the LV side characteristic voltages of nodes 5, 9, and 11 are known, and further judgment of the fault segment is required. The fault determination index C of the segment is defined as follows:

$$C = \left| \frac{\Delta U}{Z_{\text{segment}}} \right| \tag{4}$$

where $\Delta U$ denotes the characteristic voltage phasor difference between the two ends of the segment and $Z_{\text{segment}}$ denotes the line impedance of the segment. At this time, fault determination indicators of the three segments need to be calculated. Both segments 1 and 2 are located upstream of the fault point, while segment 3 contains both upstream and downstream of the fault. Therefore, indicators C of segments 1 and 2 should be approximately equal and much greater than that of segment 3. Fault can be found within segment 3 through the difference of index C.

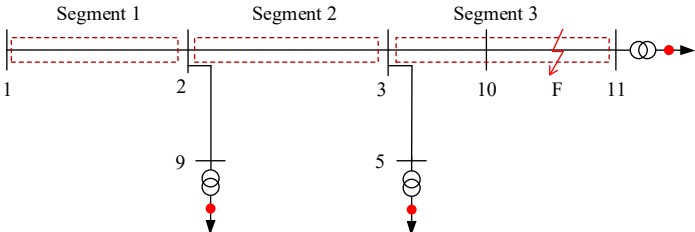

**Figure 3.** Suspected fault path for fault F.

In Figure 3, the voltage measured at node 1 is the MV side voltage and there are no measuring points of the LV side at node 2 and node 3. The LV data of these three nodes cannot be obtained directly, so the following processing is required. The characteristic voltage of the LV side at node 1 can be converted directly according to (1) and (2). The characteristic voltage at node 2 and node 3 can be calculated with the voltage of the measuring points at the end of their corresponding branches. The voltage at node 2 is calculated based on the voltage at node 9, and node 3 is calculated based on the voltage at node 5 (or node 14). It should be noted that the characteristic current flowing on the non-fault path is extremely low under the fault condition. The negative sequence current of the non-fault branch during a single-phase ground fault is almost zero. Similarly, the load current of the non-fault branch during a phase-to-phase fault is significantly lower than the short circuit current of the fault path. Therefore, the fault segment can be further determined by comparing the C-values of each segment in the suspected fault path.

### 3.5. Measuring Point Configuration

In this study, the fault segment is determined according to the principle that the characteristic voltage maximizes the value at the fault point. Theoretically, the more measuring points, the higher the accuracy of the segment location. However, the distribution network structure is complex with numerous nodes, so a great number of measuring points will lead to a significant increase in cost. To this end, the following principles of the measuring point configuration for the fault location based on the LV side voltage measurement are determined by considering the characteristics of the location method.

1. The secondary side of the distribution transformer at the end of the long branch needs to be configured with measuring points. There is no need to allocate measuring points for the short branch under the condition of meeting the positioning accuracy. As shown in Figure 2, nodes 5, 9, 11, and 14 are at the end of the long branch. However, the branch at the end of node 12 is too short to install measuring points.
2. For a feeder without branches in a larger area, the measuring point should be appropriately configured on the secondary side of the distribution transformer directly connected to the main line to meet the location accuracy. For example, the line between nodes 2 and 9 in Figure 2 does not have any branch, and the secondary side of the transformer that is directly connected to node 7 can be configured with measuring points to improve the location accuracy.

### 3.6. Flowchart of the Proposed Location Method

The process of locating the fault segment for the MV distribution networks based on the characteristic voltage of the LV side is shown in Figure 4 with the following steps:

1. Determination of the characteristic voltage. In this study, the scheme proposed in [35] is used to identify the distribution network fault types. The characteristic voltage is selected according to the principles listed in Table 3.
2. Determination of the suspected fault path. Calculate the characteristic voltage value of each measuring point and determine the measuring point where the most characteristic voltage is located. The suspicious fault path is the shortest path between the source end and this measuring point, or the shortest path between the source end and the common nodes of multiple measuring points.
3. Executing the fault segment search algorithm to determine the fault segment. The suspicious fault path is divided into segments by its branch measuring points, which should be numbered from the source side of the system. The characteristic voltage phasor difference between the two ends of the segments is used to calculate the fault determination index C of each segment. The fault segment was determined by comparing the corresponding C values of each segment.

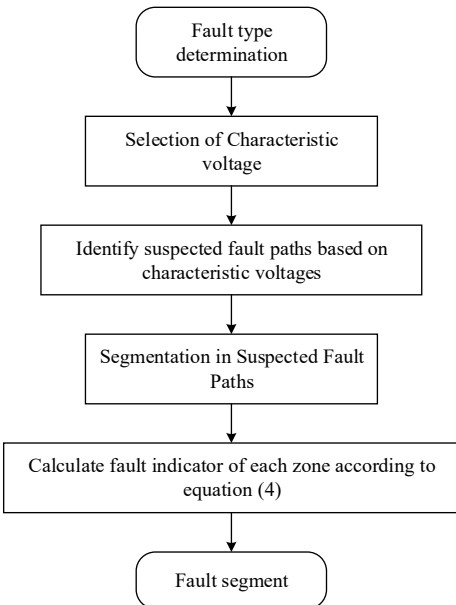

**Figure 4.** Flowchart of fault segment location.

## 4. Simulation Results

### 4.1. Case 1

The proposed scheme was validated using a modified IEEE34 node distribution network model built on the PSCAD/EMTDC platform. Combined with the actual operation of the domestic distribution network, this study only considers the three-phase network lines, and the influence of single-phase lines on the proposed method will be further discussed in future work. In addition, all loads in the simulation model are equipped with distribution transformers, which are not drawn in the figure for the sake of simplicity. The system structure is shown in Figure 5 with the voltage level of 10 kV. Based on the analysis in Section 3.5, it is known that the voltage measurement devices are configured at the reference power outlet of the distribution network as well as at the LV side of the end node of the network. The position of each LV side measuring point is shown in Figure 5.

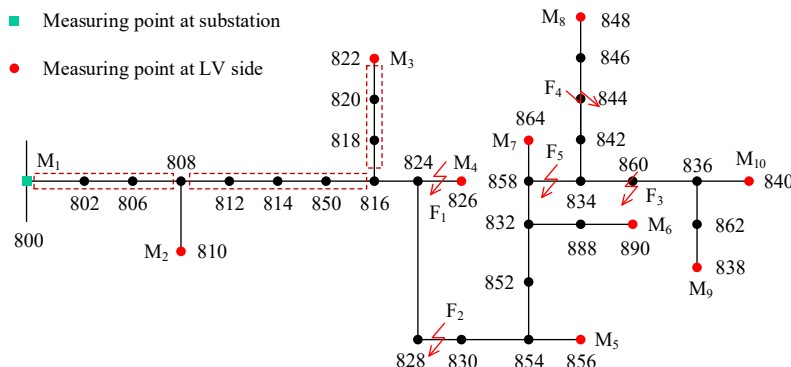

**Figure 5.** IEEE34-node distribution network.

The occurrence of single-phase metallic ground fault at node 820 is considered as an example to describe the fault segment location. Following the occurrence of the fault, three phase voltage values of each measuring point were used to calculate the negative sequence voltage value of each measuring point, which are listed in Table 4 (bold indicates the highest or lowest value). The negative sequence component of the third measuring point is the largest, determining the suspected fault path as shown in the dashed box in Figure 5. The suspected fault path was then divided into three subsections according to the branches containing the measuring point along the path.

**Table 4.** Characteristic voltage values in case 1.

| Measuring Point | Characteristic Voltage (V) | Measuring Point | Characteristic Voltage (V) |
|---|---|---|---|
| $M_1$ | 0.15 | $M_6$ | 27.76 |
| $M_2$ | 9.86 | $M_7$ | 27.74 |
| $M_3$ | 40.43 | $M_8$ | 27.73 |
| $M_4$ | 27.69 | $M_9$ | 27.73 |
| $M_5$ | 27.78 | $M_{10}$ | 27.73 |

The corresponding fault indicator C was calculated independently for each segment and all the C values were displayed in the same dimension. This aims to clearly distinguish the faulty segment from the others. As shown in Figure 6, the green markers with larger amplitude are the indicators of the fault upstream segment 808–816 and 800–808. The red marker with the smallest amplitude is the indicator between nodes 816 and 822, which is the fault segment and corresponds to the preset fault position. It should be noted that in some cases, there may be no segment that is completely downstream of the fault in the suspected fault path. Thus, only the C indicators of the upstream and fault segments exist.

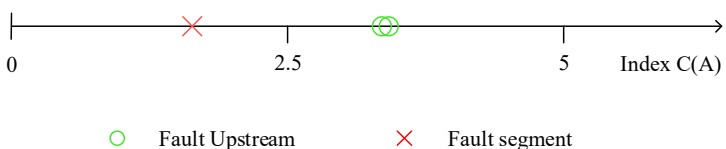

**Figure 6.** Result of single-phase grounding fault in case 1.

### 4.1.1. Effect of Fault Resistance

The fault resistance affects the magnitude of the negative sequence component in the single-phase ground fault, consequently influencing the positioning effect of single-phase ground fault. For the phase-to-phase, two-phase grounding, and three-phase grounding fault, the corresponding phase voltage at LV side of the fault point is always zero. Namely, the fault resistance does not affect the distribution of the corresponding phase voltage on the LV side. The line voltage distribution of the two faulty phases on the MV side always decreases from the rated value at substation to zero at the fault point. Its distribution is only related to the fault line. Therefore, only the effect of the fault resistance on the single-phase grounding location is discussed in this subsection. The occurrence of a single-phase ground fault at node 820 is used as an example of the fault segment location under different fault resistances. Corresponding characteristic voltage values and location results are presented in Table 5 and Figure 7, respectively.

**Table 5.** Characteristic voltage under different fault resistance in case 1.

| Measuring Point | Characteristic Voltage (V) | | | | |
|---|---|---|---|---|---|
| | $R_f = 0\ \Omega$ | $R_f = 10\ \Omega$ | $R_f = 50\ \Omega$ | $R_f = 100\ \Omega$ | $R_f = 500\ \Omega$ |
| $M_1$ | 0.15 | 0.12 | 0.07 | 0.04 | 0.01 |
| $M_2$ | 9.86 | 8.28 | 4.53 | 2.78 | 0.66 |
| $M_3$ | 40.43 | 33.97 | 18.6 | 11.41 | 2.71 |
| $M_4$ | 27.69 | 23.27 | 12.74 | 7.81 | 1.85 |
| $M_5$ | 27.78 | 23.34 | 12.78 | 7.84 | 1.86 |
| $M_6$ | 27.76 | 23.32 | 12.77 | 7.83 | 1.86 |
| $M_7$ | 27.74 | 23.31 | 12.76 | 7.83 | 1.86 |
| $M_8$ | 27.73 | 23.3 | 12.75 | 7.82 | 1.86 |
| $M_9$ | 27.73 | 23.3 | 12.76 | 7.82 | 1.86 |
| $M_{10}$ | 27.73 | 23.3 | 12.76 | 7.82 | 1.86 |

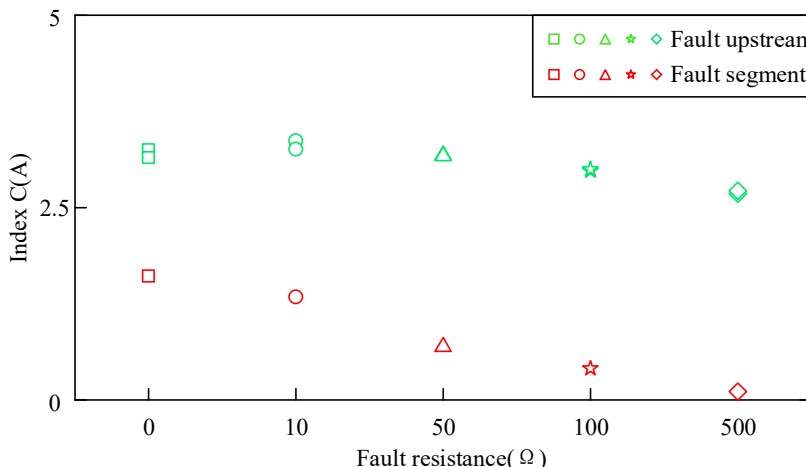

**Figure 7.** Results of under different fault resistances in case 1.

According to the location results in Figure 7, the fault with a high fault resistance of 500 Ω can still be accurately located according to index C. Among the grounding faults with fault resistance less than 3 kΩ, the percentage of fault resistance less than 140 Ω is 85% in line with relative statistics. Thus, the method can be applied to locate most single-phase grounding faults in small resistance systems. Excluding the single-phase to ground fault location, which is less affected by fault resistance, the location results of the remaining type of faults are not affected by the fault resistance.

### 4.1.2. Effect of Fault Position

In order to verify the accuracy and effectiveness of the proposed method in the entire network, simulations of a two-phase to ground fault were conducted with zero-fault resistance at different positions. The corresponding characteristic voltage values and location results are shown in Table 6 and Figure 8, respectively. The fault positions are listed as follows.

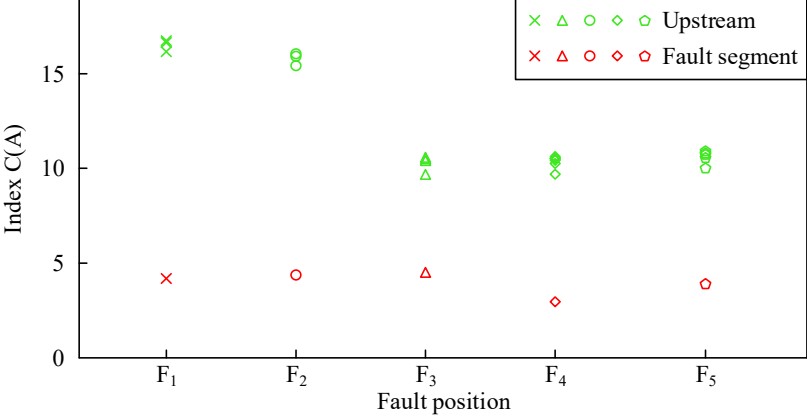

**Figure 8.** Results under different fault locations in case 1.

$F_1$: between nodes 824 and 826, 0.9 km from node 824.
$F_2$: between nodes 828 and 830, 4 km from node 828.
$F_3$: Node 860.
$F_4$: Node 844.
$F_5$: Between node 858 and 834, 1.5 km from node 858.

**Table 6.** Characteristic voltage of under different fault locations in case 1.

| Measuring Point | Characteristic Voltage (V) | | | | |
|:---:|:---:|:---:|:---:|:---:|:---:|
| | $F_1$ | $F_2$ | $F_3$ | $F_4$ | $F_5$ |
| $M_1$ | 218.4 | 218.49 | 218.79 | 218.8 | 218.78 |
| $M_2$ | 151.04 | 157.44 | 178.1 | 178.03 | 176.67 |
| $M_3$ | 24.88 | 42.95 | 98.49 | 101.03 | 97.21 |
| $M_4$ | 0 | 25.22 | 86.03 | 88.49 | 84.32 |
| $M_5$ | 5.57 | 0 | 59.90 | 62.82 | 57.89 |
| $M_6$ | 5.57 | 0 | 15.4 | 19.02 | 12.82 |
| $M_7$ | 5.56 | 0 | 9.48 | 13.2 | 6.82 |
| $M_8$ | 5.56 | 0 | 2.44 | 0 | 0 |
| $M_9$ | 5.56 | 0 | 0 | 6.26 | 0 |
| $M_{10}$ | 5.56 | 0 | 0 | 6.26 | 0 |

Based on the characteristic voltage values in Table 6, the characteristic voltage of several measuring points downstream of the fault will reach the minimum value when the fault is in the main feeder. Only the characteristic voltage of the branch measuring point reaches the minimum value when the fault is in the branch. Namely, the number of measuring points with the highest or lowest value of the characteristic voltage is related to the fault position. From the positioning results in Figure 8, the scheme can accurately distinguish the faulted and non-faulted segments at each of the fault cases. The results are consistent with the preset fault positions. Thus, the method can achieve the segment location for each fault position of the network.

### 4.1.3. Effect of Fault Type

To investigate the applicability of the proposed method to all types of faults, simulations of different types of faults are performed between nodes 818–820 with the fault resistance set to zero. Based on the localization results shown in Figure 9, the proposed method is applicable for locating all types of short-circuit faults in the small-resistance grounding system.

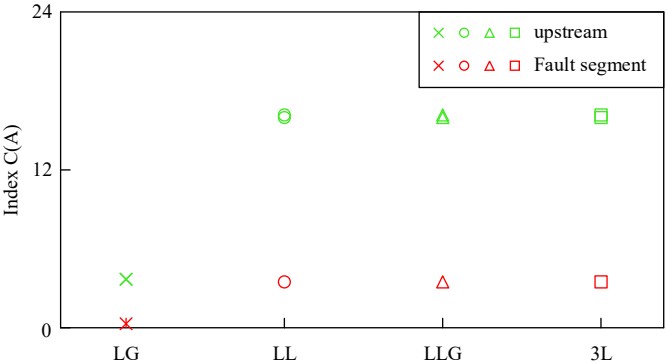

**Figure 9.** Results under different fault types in case 1.

### 4.1.4. Effect of System Grounding Mode

The system grounding mode affects the range of fault voltage variation and even determines the presence of this component. If the line to ground capacitance is ignored in an ungrounded system, the absence of negative and zero sequence components on the LV side during a single-phase fault will impact the location results. Accordingly, fault simulations with different fault types and different fault resistances are carried out in different grounding mode systems to verify the adaptability of the proposed location scheme to each grounding system. It is assumed that $F_m$ is located between nodes 824–826 and $F_n$ is located between nodes 854–856. Figure 10 presents the fault location results for different grounding modes, where the C values calculated from the characteristic

voltages are presented. The fault parameter axis indicates different fault conditions and the corresponding fault resistances are shown in parentheses.

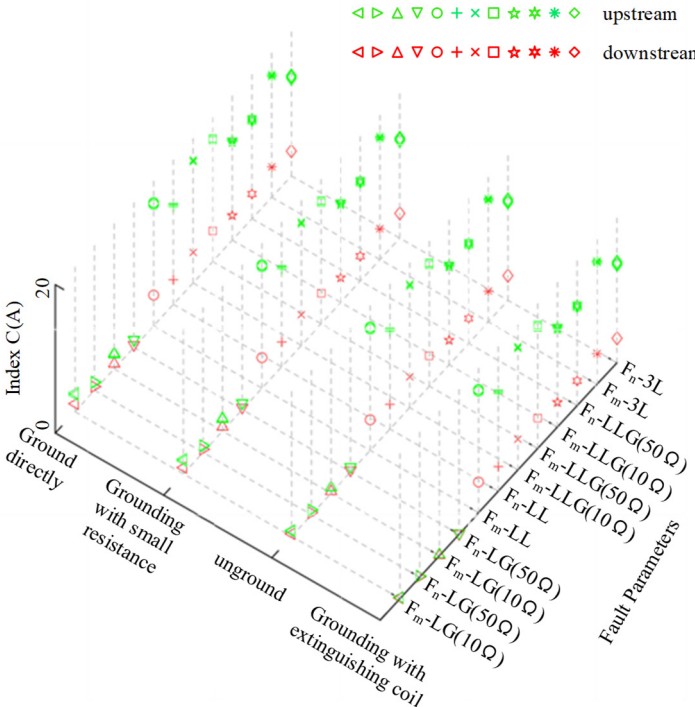

**Figure 10.** Results under different grounding modes in case 1.

Based on the location results in Figure 10, it is known that the system grounding mode does not cause any influence on the location results for two-phase faults, two-phase ground faults, and three-phase ground faults. This is determined by the amplitude distribution of the line voltage between the two faulted phases from the substation to the fault point, which is only related to the fault distance. For the single-phase fault, the negative sequence component decreases with the increase in zero-sequence impedance in the system. The location performance is slightly different in different grounding modes. In this regard, an auxiliary resistor can be connected in parallel on the arc extinguishing coil for the resonant grounding system with large zero-sequence impedance. The auxiliary resistor can be selected to be put in after the fault to increase the fault current and improve the single-phase ground fault location accuracy of this method.

### 4.1.5. Effect of Equipment Measurement Error

A measurement device generally introduces an error of less than 0.5% due to environmental interference and the measurement error of the device itself. The presence of lateral branches in the distribution network results in the formation of a larger number of T-nodes. When the fault location is close to the T-node upstream of it, the characteristic voltage magnitudes at the two terminals of the fault section will also be identical. The measurement error of the device may lead to the incorrect determination of the maximum characteristic voltage measuring point, leading to locating the fault in the adjacent section.

Considering the single-phase ground fault as an example, the fault resistance is set to 50 Ω. A measurement error of −0.5% to +0.5% is randomly introduced to each measuring point to demonstrate the robustness of the location method to measurement errors. According to the previous analysis, when a fault is close to the T-node, it may result in the characteristic voltage measurement at the first end of the faulted section being larger than the end measurement. The distance of the fault point from the T-node when they are equal is defined as the critical distance of the node. The smaller its value, the better the location performance. Simulation data demonstrate that the critical distance of most

T-nodes is zero, and only a few nodes are affected by faults near the downstream. All these critical distances are within 30m, which is an acceptable range. In general, the fault location scheme proposed in this study can accurately achieve segment location in the presence of measurement errors.

4.1.6. Sensitivity Analysis

The above sections discuss the influence of the system, fault parameters, and measurement conditions on the location method in this paper, which will be summarized in this section. Simulation results show that the proposed method can locate the single-phase ground fault section with a fault resistance up to 500 $\Omega$. Single-phase to ground fault location is a difficult problem for small current grounding systems. The method in this paper can still achieve a fault section location in general, but the overall effect is not as good as other systems. Future work will focus on overcoming this problem. Examples of different fault locations show that the proposed method is not affected by the fault location and can realize the fault location of the whole network under the configuration of the proposed measurement points. In this paper, the characteristic voltage used for fault segment location is determined adaptively according to different fault types, which can realize all asymmetric faults and symmetric fault segment locations. Under the error of the existing measuring devices, this method can still accurately locate the fault area by using the characteristic voltage of the low voltage side. When the fault position is extremely close to the T-node, it may cause the fault to lock to an adjacent segment of the real faulted segment. The distance of the fault from the T-node when a misclassification is just generated is called the critical distance, which is determined by the accuracy of the measurement terminal. In this study, after adding appropriate errors in the validation process, the misclassification may occur only when the fault location is less than 30m away from the T-node, and such a result is quite satisfactory.

4.1.7. Comparison with Existing MV Side Methods

A comparison between the proposed method and the existing typical fault location methods is given in Table 7. The proposed method in this paper has advantages in terms of voltage measurement level, applicable fault type, and fault resistance adaptability. Most other methods need to obtain the voltage and current information on the MV side, requiring the addition of transformers. The method proposed in this study can be combined with existing intelligent fusion terminals, electricity information collection systems, distribution station house terminals, and smart meters for voltage acquisition. It does not require a strict synchronization measurement, which significantly reduces hardware costs and is easier to implement.

**Table 7.** Comparison with existing methods.

| Reference | Measured Voltage | Fault Types | Synchronizition | Resistance |
|-----------|------------------|-------------|-----------------|------------|
| This paper | 0.4 kV | all | No | <500 $\Omega$ |
| [26] | 13.8 kV | all | Yes | <100 $\Omega$ |
| [29] | 13.8 kV | LG, 3L | Yes | <10 $\Omega$ |
| [34] | 13.8 kV | all | Yes | <100 $\Omega$ |
| [35] | 20 kV | LLG, 3L | Yes | <100 $\Omega$ |
| [36] | 20kV | LG, LLG, 3L | Yes | <100 $\Omega$ |
| [37] | 24.9kV | LG | Yes | <500 $\Omega$ |

*4.2. Case 2*

Figure 11 shows the actual distribution network model with 48 nodes in some areas, which operates at a voltage level of 10 kV and is grounded by a low resistor. According to the positioning principle in this paper, combined with the accuracy and economy of the positioning scheme, monitoring points are configured on the LV side of the nodes indicated in red in Figure 11, including a total of 10 points at the substation end.

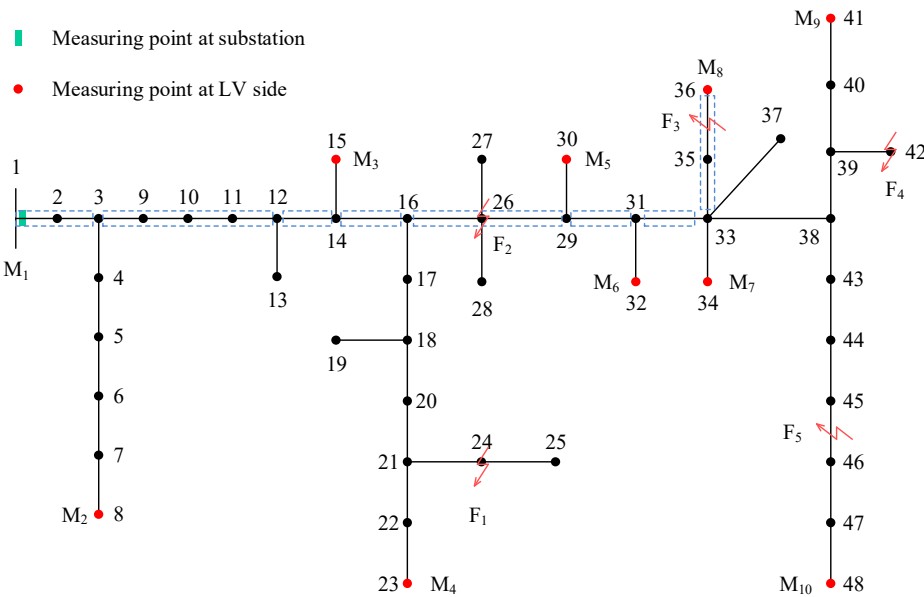

**Figure 11.** Distribution network of 48-node in case 2.

To further evaluate the effectiveness and adaptability of the location scheme proposed in this study, several fault cases are simulated by considering different fault resistances, different fault locations, and different faults. Moreover, random measurement errors within 0.5% are also introduced into these cases. The characteristic voltage values of a single-phase ground fault and a two-phase ground fault are shown in Tables 8 and 9, respectively. The corresponding final location results are shown in Figures 12 and 13. Since the distribution characteristics of the corresponding characteristic voltage (phase voltage) magnitude on the LV side are identical at the same fault location among the two-phase ground fault, phase-to-phase fault, and three-phase fault, only the characteristic voltage data for the two-phase ground fault are given below. The data for the remaining two types of faults are consistent with this. The specific fault locations are as follows:

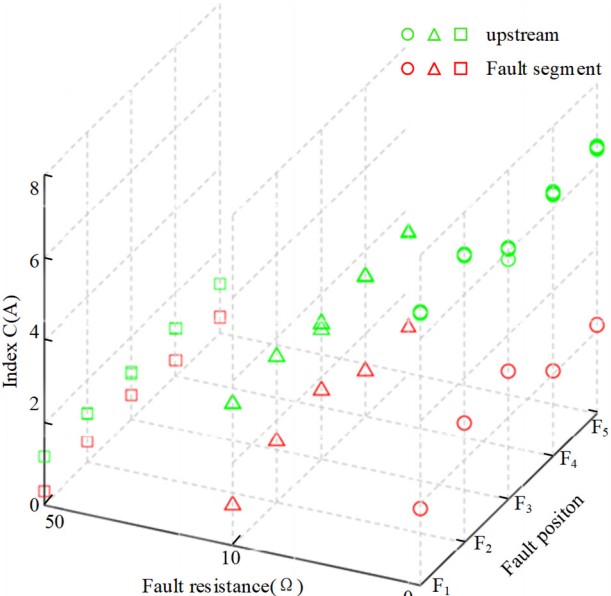

**Figure 12.** Results of single-phase ground fault in case 2.

**Table 8.** Characteristic voltage of single-phase fault in case 2.

| Measuring Point | Characteristic Voltage (V) (R$_f$ = 0 Ω) | | | | | Characteristic Voltage (V) (R$_f$ = 10 Ω) | | | | | Characteristic Voltage (V) (R$_f$ = 50 Ω) | | | | |
|---|---|---|---|---|---|---|---|---|---|---|---|---|---|---|---|
| | F$_1$ | F$_2$ | F$_3$ | F$_4$ | F$_5$ | F$_1$ | F$_2$ | F$_3$ | F$_4$ | F$_5$ | F$_1$ | F$_2$ | F$_3$ | F$_4$ | F$_5$ |
| M$_1$ | 0.06 | 0.06 | 0.06 | 0.06 | 0.06 | 0.03 | 0.03 | 0.03 | 0.03 | 0.03 | 0.01 | 0.01 | 0.01 | 0.01 | 0.01 |
| M$_2$ | 1.12 | 1.13 | 1.06 | 1.06 | 1.06 | 0.59 | 0.59 | 0.57 | 0.57 | 0.57 | 0.2 | 0.2 | 0.2 | 0.2 | 0.2 |
| M$_3$ | 3.04 | 3.06 | 2.88 | 2.89 | 2.89 | 1.6 | 1.61 | 1.56 | 1.56 | 1.55 | 0.55 | 0.55 | 0.54 | 0.54 | 0.54 |
| M$_4$ | 10.17 | 9.45 | 8.89 | 8.93 | 8.9 | 5.36 | 4.96 | 4.81 | 4.8 | 4.79 | 1.84 | 1.7 | 1.67 | 1.67 | 1.66 |
| M$_5$ | 9.38 | 9.76 | 9.62 | 9.65 | 9.63 | 4.94 | 5.12 | 5.2 | 5.19 | 5.18 | 1.7 | 1.75 | 1.81 | 1.8 | 1.8 |
| M$_6$ | 9.36 | 9.73 | 10.59 | 10.62 | 10.6 | 4.93 | 5.11 | 5.72 | 5.71 | 5.71 | 1.69 | 1.75 | 1.99 | 1.98 | 1.98 |
| M$_7$ | 9.35 | 9.73 | 10.92 | 10.96 | 10.94 | 4.92 | 5.1 | 5.91 | 5.9 | 5.89 | 1.69 | 1.75 | 2.05 | 2.05 | 2.04 |
| M$_8$ | 9.35 | 9.72 | 13.51 | 10.96 | 10.94 | 4.92 | 5.1 | 7.3 | 5.9 | 5.89 | 1.69 | 1.75 | 2.54 | 2.05 | 2.04 |
| M$_9$ | 9.32 | 9.7 | 10.9 | 12.77 | 12.3 | 4.91 | 5.09 | 5.89 | 6.87 | 6.62 | 1.69 | 1.74 | 2.05 | 2.38 | 2.3 |
| M$_{10}$ | 9.31 | 9.69 | 10.88 | 12.32 | 12.87 | 4.91 | 5.09 | 5.88 | 6.63 | 6.93 | 1.68 | 1.74 | 2.05 | 2.3 | 2.41 |

**Table 9.** Characteristic voltage of two-phase ground fault in case 2.

| Measuring Point | Characteristic Voltage (V) | | | | |
|---|---|---|---|---|---|
| | F$_1$ | F$_2$ | F$_3$ | F$_4$ | F$_5$ |
| M$_1$ | 221.25 | 220.25 | 222.16 | 221.88 | 221.83 |
| M$_2$ | 182.27 | 166.28 | 199.09 | 194.71 | 194.07 |
| M$_3$ | 116.65 | 69.03 | 159.72 | 146.81 | 144.82 |
| M$_4$ | 15.69 | 17.14 | 139.76 | 121.50 | 118.52 |
| M$_5$ | 87.30 | 0 | 123.44 | 101.30 | 97.81 |
| M$_6$ | 87.08 | 0 | 102.12 | 73.90 | 69.50 |
| M$_7$ | 87.04 | 0 | 95.18 | 64.51 | 59.66 |
| M$_8$ | 86.82 | 0 | 0 | 64.35 | 59.52 |
| M$_9$ | 86.76 | 0 | 94.88 | 1.49 | 20.61 |
| M$_{10}$ | 86.66 | 0 | 94.78 | 29.24 | 0 |

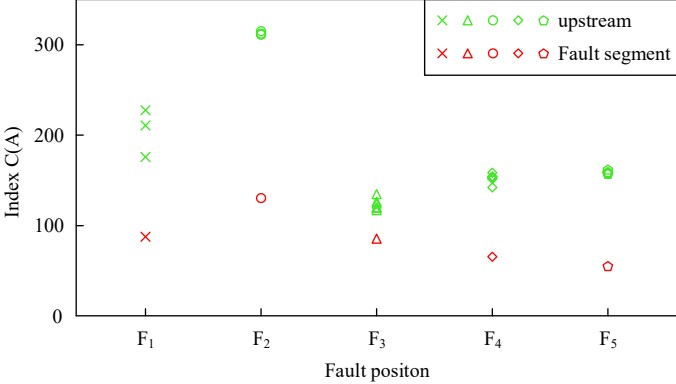

**Figure 13.** Results of two-phase ground fault in case 2.

F1: Node 24.
F2: Node 26.
F3: Between nodes 35 and 36, 900 m from node 35.
F4: Node 42.
F5: Between nodes 45 and 46, 91 m from node 45.

It can be learned from the single-phase ground fault location results in Figure 12 that the indicator C decreases with the increase in fault resistance under the same fault location. The increase in fault resistance will reduce the voltage drop of the negative sequence voltage on the line. In the two-phase fault, two-phase ground fault, and three-phase ground fault, the amplitude distribution of the LV side characteristic voltage (phase voltage) related to the two MV side fault phases are not affected by the fault resistance or fault type. Its

distribution is solely determined by the fault location. Therefore, Table 9 and Figure 13 only provide the characteristic voltages and location results for two-phase ground fault with zero fault resistance as a representative case.

Therefore, the proposed method can achieve accurate fault segment identification for different fault locations and fault resistances. Excluding single-phase to ground fault, the location effect of the remaining types of faults is not affected by the fault resistance. This example also further verifies the adaptability of the proposed method in different distribution networks.

## 5. Conclusions

An economical and practical fault segment location method based on the characteristic voltage of the LV side is proposed in this study to resolve the high installation cost of measuring points and maintenance difficulties of the fault location methods based on the measurements of the MV side. Based on the analysis of the distribution characteristics of LV side voltage in the distribution network, the suspicious fault path determination and fault segment search method based on the LV side characteristic voltage are proposed. The following conclusions are drawn.

1. The variation characteristics of different LV side voltages vary for different types of MV side faults. The variations in the phase, sequence, and line voltages are small in single phase to ground faults. For the remaining short-circuit faults, the variations of phase voltage, sequence voltage, and line voltage are considerable.
2. The characteristic voltage of single-phase to ground fault reaches the maximum value at the fault point while that of the remaining types of faults reaches the minimum value at the fault point. Downstream of the fault, the characteristic voltage changes little along the line and remains almost the same as the fault point. Moreover, a method for determining the suspected fault path based on the characteristic voltage of the LV side is proposed.
3. The characteristic voltage drops per unit distance upstream, and the fault section and the downstream fault are different, which can be used to search for the fault section in the suspected fault path. The correctness of the method proposed in this study is verified by numbers of simulation cases of the IEEE34 network and the actual distribution network in a region.
4. The fault segment location scheme based on the characteristic voltage on the LV side can realize the location of the fault segment of the MV line without the measurement information on the MV side of the distribution network. The method is based on the distributed voltage measurement for the fault location, and the existing LV measurement terminals in the distribution network, such as the power information collection system and the intelligent fusion terminal, can realize the collection of the voltage and power information. In terms of technology, the method can determine the fault path by using only the distribution characteristics of the characteristic voltage on the low-voltage side and it can identify the fault section by using the characteristic voltage drop per unit of the line without particularly complicated mathematical calculations, and the steps are simple. In terms of location cost, the fault location can be performed based on the voltage data collected by the existing LV measurement terminals in the distribution network without the need for additional measurement terminals, which greatly reduces the hardware cost for the location. This method is of great value for distribution networks lacking medium voltage measurement terminals, especially in rural areas.
5. In future work, we will focus on how to improve the accuracy of the single-phase ground fault location in low-current grounding systems, and the effect of single-phase lines on the proposed method will be further discussed.

**Author Contributions:** Conceptualization, methodology, formal analysis, writing—original draft preparation, D.Z.; supervision, W.Z.; resources, software, C.W.; project administration, X.X. All authors have read and agreed to the published version of the manuscript.

**Funding:** This work was supported by the National Key Research and Development Program of China [grant number 2020YFF0305800] and the Natural Science Foundation of Sichuan [grant number 2022NSFSC0234].

**Data Availability Statement:** The data can be provided when required.

**Conflicts of Interest:** The authors declare no conflict of interest.

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
