# Peer review of "Fault Segment Location for MV Distribution System Based on the Characteristic Voltage of LV Side"

_electronics, doi:10.3390/electronics12071734_

Round 1

Reviewer 1 Report

In this paper, a fault location method for the MV distribution network based on the characteristic voltage at the LV side is proposed. The characteristic voltage is selected adaptively according to the fault type. The simulation of the IEEE 34 system built in PSCAD/EMTDC and the real 23 distribution network.  The work is interesting. Authors are asked to consider the following issues: 

1. Avoid the repeated usage of voltage in the title. You can use LV, MV ...

2. Update the abstract with the assessment of the proposed voltage-based fault location compared with the previous method. 

3. Literature review is limited. Collect similar studies with critical issues for each. The research gap and contribution must be clear. 

4. Applications need more discussion. 

5. The configuration of 34-bus must include transformers, single and three phases, and study their impacts on the fault location procedure. 

6. Add the sensitivity analysis and failure rate of the proposed scheme.

7. Add the future trend compared to the existing work.

Author Response

Point 1: Avoid the repeated usage of voltage in the title. You can use LV, MV ...

Response 1: We sincerely thank the reviewer for careful reading. As suggested by the reviewer, we have corrected the “medium-voltage” into “MV”, the “low-voltage” into “LV” in the title. These changes are marked in red in the title in the revised manuscript.

Point 2: Update the abstract with the assessment of the proposed voltage-based fault location compared with the previous method.

Response 2: Thanks for the reviewer's suggestion. We agree that it is necessary to assess the proposed fault location method in comparison with previous methods. Due to the large content, it is inconvenient to supplement in the abstract. Hence, relevant discussion is added at the end of paragraph 5 of the introduction. All changes are marked in red in the revised manuscript.

Point 3: Literature review is limited. Collect similar studies with critical issues for each. The research gap and contribution must be clear.

Response 3: We think this is an excellent suggestion. We have replaced some of the literature based on MV side measurement in paragraph 3 of the introduction and explained their limitations at the beginning of paragraph 4, which leads to the advantages of the LV-measurement based location method. We have added the literature based on LV-side measurement in paragraph 4 of the introduction, further completing the existing summary of LV-side fault location methods. The shortcomings of the existing LV-side based methods are described at the end of paragraph 4. In addition, the main contributions of this study are listed in points in the last part of the introduction. All changes are marked in red in the revised manuscript.

Point 4: Applications need more discussion.

Response 4: Thanks for the reviewer's suggestion. In the fourth part of the conclusion, we have further discussed the combination of the proposed method with existing LV measuring terminals and the application scenarios. All changes are marked in red in the revised manuscript.

Point 5: The configuration of 34-bus must include transformers, single and three phases, and study their impacts on the fault location procedure.

Response 5: As stated by the reviewer, 34-bus network contains regulators, distribution transformers and the lines involve a mixture of single-phase and three-phase scenarios. Considering the actual operation of the domestic distribution network, the parameters of node 34 have been suitably modified and we have only considered three-phase lines. In fact, the presence of the regulator does not have an impact on the positioning method proposed in this study. We have therefore removed the regulators for the sake of simplicity of the diagram. In addition, all loads are equipped with distribution transformers, which have been omitted from the legend. In future work we will further discuss the impact of the presence of single-phase lines on the proposed method for this study. The relevant notes are also added in the first paragraph of the simulation results, which are marked in red in the revised manuscript.

Point 6: Add the sensitivity analysis and failure rate of the proposed scheme.

Response 6: We thank the reviewer for pointing out this issue. We complement the sensitivity analysis of the method in this paper in subsection 4.1.6 of the simulation validation. The misjudgment of fault segment and the corresponding critical distance due to the proximity of the defect to the T-junction are discussed. All arguments are marked in red in the revised manuscript.

Point 7: Add the future trend compared to the existing work.

Response 7: Thanks for the reviewer's suggestion. Our future work will be focused on improving the location accuracy of single-phase to ground fault in low-current grounding systems and analyzing the effect of single-phase lines on the proposed method. These arguments have been added to the last paragraph of the conclusion in the revised manuscript.

Thanks again for the reviewer's suggestion. In view of the quality of the article's English, we have asked a professional organization to polish and revise it. We have tried our best to improve the quality of the language in the article and marked in blue all the modified words and sentences. All amendments do not change the meaning of the original text.

Reviewer 2 Report

Scientific work related to ensuring the improvement of reliability, safety, fault tolerance of electric power systems is currently very relevant. This article is no exception. It considers the issue of developing an economical and practical method for localizing fault segments based on the characteristic voltage on the low voltage side.

All calculation schemes and algorithms were tested using a modified IEEE34 node distribution network model built on the PSCAD/EMTDC platform.

The following important results were obtained during the study:

- the characteristics of the change of different voltages on the low voltage side are different for different types of faults on the medium voltage side;

- the characteristic voltage of a single-phase earth fault reaches its maximum value at the point of the fault, and for other types of faults it reaches the minimum value at the point of the fault;

- characteristic voltage drops per unit distance in the ascending, damaged and descending sections of the damage site are different;

- the fault location scheme based on the characteristic voltage on the low voltage side can realize fault location of the medium voltage line without measurement information on the medium voltage side of the distribution network.

There are several wishes (recommendations) for the authors:

- in the list of references, 50% of sources are older than 5 years, which can be reduced based on the relevance and a large number of similar works in recent times;

- since the conclusion indicates the economic attractiveness of the method proposed by the authors, it would be nice to see a small technical and economic assessment, or not to indicate this fact without voice.

In general, the work is a complete scientific study on a topical topic and has novelty. Deserves to be published after minor modifications.

Author Response

Point 1: in the list of references, 50% of sources are older than 5 years, which can be reduced based on the relevance and a large number of similar works in recent times;

Response 1: It’s a great suggestion. We have replaced the older references with ones that are less than 5 years old. All new added references are marked in red in the revised manuscript.

Point 2: since the conclusion indicates the economic attractiveness of the method proposed by the authors, it would be nice to see a small technical and economic assessment, or not to indicate this fact without voice.

Response 2: Thanks for the reviewer's suggestion. The proposed method has advantages over other methods in terms of technical realization and economic cost. Relevant discussions are added in the fourth part of the conclusion. All changes are marked in red in the revised manuscript.

Thanks again for the reviewer's suggestion. In view of the quality of the article's English, we have asked a professional organization to polish and revise it. We have tried our best to improve the quality of the language in the article and marked in blue all the modified words and sentences. All amendments do not change the meaning of the original text.

Round 2

Reviewer 1 Report

The paper is improved considering my issues.